# Shedding of the Endothelial Glycocalyx Independent of Systemic Tryptase Release during Oncologic Oral Surgery: An Observational Study

**DOI:** 10.3390/jcm11195797

**Published:** 2022-09-30

**Authors:** Hendrik Drinhaus, Daniel C. Schroeder, Nicolas Hunzelmann, Holger Herff, Thorsten Annecke, Bernd W. Böttiger, Wolfgang A. Wetsch

**Affiliations:** 1University of Cologne, Faculty of Medicine, and University Hospital of Cologne, Department of Anaesthesiology and Intensive Care Medicine, 50937 Cologne, Germany; 2German Armed Forces Central Hospital of Koblenz, Department of Anaesthesiology and Intensive Care, 56072 Koblenz, Germany; 3University of Cologne, Faculty of Medicine, and University Hospital of Cologne, Department of Dermatology, 50937 Cologne, Germany; 4University of Witten/Herdecke, Kliniken der Stadt Köln gGmbH, Department of Anaesthesiology and Intensive Care Medicine, 51109 Cologne, Germany

**Keywords:** oral surgery, head and neck cancer, glycocalyx, tryptase

## Abstract

The endothelial glycocalyx and endothelial surface layer are crucial for several functions of the vasculature. Damage to the glycocalyx (“shedding”) occurs during diverse clinical conditions, including major surgery. Mast cell tryptase has been proposed as one possible “sheddase”. During oncologic oral surgery, glycocalyx shedding could be detrimental due to loss of vascular barrier function and consequent oedema in the musculocutaneous flap graft. Concentrations of the glycocalyx components heparan sulphate and syndecan-1, as well as of tryptase in blood serum before and after surgery, were measured in 16 patients undergoing oncologic oral surgery. Secondary measures were the concentrations of these substances on postoperative days 1 and 2. Heparan sulphate rose from 692 (median, interquartile range: 535–845) to 810 (638–963) ng/mL during surgery. Syndecan-1 increased from 35 (22–77) ng/mL to 138 (71–192) ng/mL. Tryptase remained virtually unchanged with 4.2 (3–5.6) before and 4.2 (2.5–5.5) ng/mL after surgery. Concentrations of heparan sulphate and syndecan-1 in serum increased during surgery, indicating glycocalyx shedding. Tryptase concentration remained equal, suggesting other sheddases than systemic tryptase release to be responsible for damage to the glycocalyx. Investigating strategies to protect the glycocalyx during oncologic oral surgery might hold potential to improve flap viability and patient outcome.

## 1. Introduction

The endothelial glycocalyx is a fine layer of glycoproteins coating the endoluminal surface of the vasculature. Together with plasma proteins, it forms the endothelial surface layer (ESL). Syndecan-1 is a transmembrane molecule with attachment sites for glycosaminoglycans on the luminal sides of the endothelial cells. It is also involved in transmembrane signalling. Heparan sulphate is the main glycosaminoglycan of the endothelial glycocalyx and is bound to syndecan-1 within the lumen of the vessel [1]. The ESL is essential for several functions of the vessels, including vascular permeability, vascular tone and coagulation. In recent years, the degradation of the glycocalyx and the shedding of its components, including heparan sulphate and syndecan-1, into the circulation have been demonstrated in a number of pathologic conditions, including sepsis, major trauma, or cardiac arrest [2,3]. Major surgery has also been shown to be associated with shedding of the glycocalyx [2]. If the glycocalyx is shed, the functions of the ESL are disrupted.

Two conditions frequently encountered during oral cancer surgery, infusion of large volumes of fluid and ischaemia reperfusion during flap surgery, are also known contributors to glycocalyx shedding [4,5]. On the other hand, signs consistent with ESL disruption—particularly loss of vascular barrier function—frequently occur during oncologic oral surgery. This can be seen, for instance, by high fluid volumes that need to be infused perioperatively. The precise mechanisms of glycocalyx shedding and the substances involved, the so-called “sheddases”, have not been definitely elucidated yet. Among others, activation of mast cells and release of their enzyme tryptase have been suggested as possible mechanisms [6,7]. We hypothesised that tryptase release from mast cells leading to glycocalyx shedding would occur during major oncologic oral surgery, in particular involving free or pedicled flap reconstruction. As fluid overload can represent a danger to the perfusion of flaps used for reconstruction after oral cancer surgery [8,9], investigating shedding of the glycocalyx during oral surgery and developing strategies to protect it might hold potential to improve flap viability and to improve the medical and surgical perioperative course. Therefore, we set up a study to analyse glycocalyx shedding—expressed by levels of the glycocalyx components heparan sulphate and syndecan-1—and tryptase concentration in patients undergoing oral cancer surgery.

## 2. Materials and Methods

### 2.1. Ethics

Ethical approval for this study was obtained from the Ethics Committee of the Medical Faculty of the University of Cologne, Cologne, Germany (approval number 13–229). All patients provided written informed consent before enrolment.

### 2.2. Study Registration

The study was registered at ClincialTrials.gov, Identifier: NCT01921049.

### 2.3. Patient Recruitment

Based on earlier studies investigating glycocalyx shedding during surgery, we estimated that 16 patients would provide statistically reliable data for a pilot study to test our hypothesis [10]. We included adult, non-pregnant patients with malignant neoplasms within the oral cavity undergoing elective major tumour resections with local reconstruction using autologous grafts. Exclusion criteria were age <18 years, pregnancy, emergency and revision surgery, and failure to obtain informed consent. Sixteen consecutive patients meeting the inclusion criteria were enrolled for this part of the study investigating glycocalyx shedding.

### 2.4. Measurement of Glycocalyx Shedding

Blood was drawn at four time-points: (1) immediately before surgery, after induction of general anaesthesia; (2) after surgery was completed, upon admission to the intensive care unit (ICU); (3) in the morning of the first postoperative day in ICU; and (4) in the morning of the second postoperative day in ICU. Specimens were centrifuged, aliquoted and stored at −80 °C. Serum concentration of tryptase was measured by fluoroimmunoenzyme assay (ImmunoCAP Tryptase; Thermo Fisher Scientific, Uppsala, Sweden) and levels of syndecan-1 (Syndecan-1, Diaclone, Besancon, France) and heparan sulphate (Heparan sulphate, Cusabio Biotech CO., Wuhan, China) in serum were measured by enzyme-linked immunosorbent assays (ELISA), according to the manufacturer’s guidelines. In brief, standards with known concentration as supplied, or 50–100 µL blood serum samples, respectively, were applied to antibody-coated wells. Further reagents were added and washed as instructed by the manufacturer. After stopping the chromogenic reaction, adsorption of light at the specified wavelength was measured in a 96-well microplate reader (Multiskan GO, Thermo Fisher Scientific, Waltham, MA, USA). Concentrations were calculated by extrapolation from a standard curve derived from the standards supplied by the manufacturer. Due to technical issues with the ELISA measurement, values of syndecan-1 could only be obtained from 12 of the 16 patients included in the study.

### 2.5. Statistical Analysis

Data were analysed and visualised with GraphPad Prism 9.4 (GraphPad Software Inc., San Diego, CA, USA). Data are presented as median and interquartile range unless otherwise stated. We used the Shapiro–Wilk test to test for normal distribution. To compare two paired groups of non-normally distributed data, we used the Wilcoxon matched-pair signed-rank test. In cases of normally distributed data, we performed the paired *t*-test and for consistency and statistical prudence added the Wilcoxon matched-pair signed-rank test. Significance level was defined as α = 0.05.

## 3. Results

### 3.1. Demographic and Clinical Data

Demographic and basic clinical data of the patients are listed in Table 1. All patients underwent radical resection of their respective tumours. Apart from one patient, all defects were reconstructed with free-flap grafts, the harvest sites of which are listed in Table 1. Data concerning perioperative fluid balance are listed in Table 2.

### 3.2. Shedding of Glycocalyx Components

Heparan sulphate concentration in blood serum rose from 692 (interquartile range: 535–845) ng/mL before surgery to 810 (638–963) ng/mL upon admission to the intensive care unit. This increase was relatively small, but statistically significant (Wilcoxon matched-pair signed-rank test *p* = 0.024). Concentrations of heparan sulphate remained at 789 (617–913) ng/mL and 787 (627–1243) ng/mL on the first and second postoperative day (Figure 1).

There were two patients with extreme values as determined by the ROUT method with Q = 1% [11]. When excluding these outliers, the difference remained statistically significant (Wilcoxon matched-pair signed-rank test *p* = 0.005, paired *t*-test *p* = 0.005).

Syndecan-1 concentration in blood serum (*n* = 12 as explained above) rose from 35 (22–77) ng/mL before surgery to 138 (71–192) ng/mL upon admission to the ICU (Wilcoxon matched-pair signed-rank test *p* = 0.003) and receded to 123 (84–192) ng/mL on the first and 61 (48–101) ng/mL on the second postoperative day (Figure 2).

### 3.3. Concentration of Tryptase

Tryptase concentration in serum remained equal during surgery with 4.2 (3–5.6) ng/mL before surgery and 4.2 (2.5–5.5) ng/mL upon admission to the ICU (Wilcoxon matched pair signed-rank test *p* = 0.514, paired *t*-test *p* = 0.613). On the first and second postoperative days, tryptase concentration was 3.7 (2.7–5.4) ng/mL and 4.3 (2.6–6.6) ng/mL, respectively (Figure 3). The ROUT method identified one patient with extraordinarily high values of tryptase (between 16 and 20 ng/mL and hence 4–5 times the median) at all four measurements. When excluding this outlier, there were still no significant changes in the perioperative concentration of tryptase (Wilcoxon matched-pair signed-rank test *p* = 0.261, paired *t*-test *p* = 0.353).

## 4. Discussion

We confirmed our hypothesis that components of the endothelial glycocalyx are released into the systemic circulation during major oral surgery, implying shedding of the glycocalyx. As this increase in glycocalyx-shedding products did not concur with an increase in tryptase activity, shedding of the glycocalyx in this case appears to be independent of mast-cell activation and tryptase release, so this part of the study hypothesis was not confirmed.

This finding aligns with the glycocalyx shedding detected after other types of major surgery, such as abdominal surgery [12], vascular surgery [5], or cardiac surgery with or without a heart–lung machine [13,14], lung surgery or transplant surgery [10,15]. The time courses of the concentrations of syndecan-1 and heparan sulphate differ from each other. While both rise during surgery, only syndecan-1 decreased close to its preoperative level in our patients, whereas heparan sulphate remained high until the end of our measurements on the second postoperative day. This resembles the time courses observed in patients undergoing abdominal aortic repair or uncomplicated abdominal surgery [5,12,16]. By contrast, faster returns to normal levels of syndecan-1 and heparan sulphate have been measured after cardiac surgery and [5,13], on the other hand, slower returns to normal levels could be observed in patients remaining septic after abdominal surgery [12,16].

Several factors might contribute to glycocalyx shedding during oncologic oral surgery: free-flap surgery includes ischaemia and reperfusion of the graft. Ischaemia-reperfusion has been shown to cause shedding of the glycocalyx in various clinical and experimental setups [4,5,17]. It can be hypothesised that ischaemia and subsequent reperfusion in the free-flap graft lead to glycocalyx shedding in the graft itself. Additionally, if mediators causing shedding of the glycocalyx accumulate in the graft and are subsequently released into the circulation after anastomosis, the glycocalyx of the vasculature beyond the graft may be damaged, too. A further contributing factor might be infusion of larger volumes of fluid. Volume loading leads to glycocalyx shedding via the release of atrial natriuretic peptide (ANP) [4]. As a median infusion volume of 5550 mL during surgery in our patient collective demonstrates, many patients received large volumes of fluids during oncologic oral surgery. Infusion of large fluid volumes, however, might not only be a cause but also a consequence of glycocalyx shedding. As one of the main functions of the glycocalyx and the endothelial surface layer is to maintain vascular barrier function, damages to the glycocalyx lead to endothelial leakage [18]. This causes extravasation and can therefore exacerbate intravascular hypovolemia and, consequently, is often treated with high volumes of infusion to maintain sufficient mean arterial pressure for organ perfusion. Vascular hyperpermeability due to glycocalyx shedding has been demonstrated in a cell-culture model of Dengue-virus-induced glycocalyx shedding [19], a mouse model of abdominal sepsis [20], and in patients with burn injuries [21]. Apart from the general adverse effects of vascular leakage, such as generalized oedema, pulmonary oedema, or pleural effusion, its specific effect in oral flap surgery should be of concern. Capillary leakage in the area of the flap graft might contribute to tissue oedema and thereby jeopardize flap viability. While there are no studies directly investigating an association between status of the endothelial glycocalyx and flap viability, there are hints that a link between these two might exist. An older analysis of predictors of complications after major head and neck surgery (mostly oncologic surgery) has identified the infusion of large volumes of fluid as a risk factor for surgical complications, including flap failure [8]. Later trials have investigated the effect of goal-directed infusion therapy, implying the infusion of lower volumes than with conventional fluid management, on surgical and medical complications after head and neck surgery. Goal directed therapy led to higher flap survival and reduced rates of flap necrosis, respectively [22,23]. Some authors suspect the glycocalyx to be involved in the association between high infusion rates and worsened flap outcome [9].

In this study, tryptase levels did not change perioperatively. Therefore, it appears unlikely that systemic release of tryptase plays a major role as an enzyme causing shedding of the glycocalyx (so called “sheddase”) in the glycocalyx shedding observed in our patients. However, local release by perivascular tissue mast cells, which is not enough to increase systemic serum concentrations might still contribute to glycocalyx shedding. It remains to be elucidated which pathophysiological pathways and sheddases, such as matrixmetalloproteinases or heparinase, cause glycocalyx damage during oral surgery [7,24].

### Limitations

The present study is a monocentric study with a limited number of patients. We assessed glycocalyx-shedding by laboratory testing of shedding products (heparan sulphate and syndecan-1) into the circulation only. A complimentary method of measuring glycocalyx-shedding, visualization of the perfused boundary region in sublingual vessels by sidestream dark field microscopy could not be employed in our patients, as insertion of the camera probe would have interfered with the surgical field and might have endangered the intra-oral flap [25].

## 5. Conclusions

In this pilot study, we found an increase in heparan sulphate and syndecan-1 concentrations in the blood, suggesting shedding of the endothelial glycocalyx during major oncologic oral surgery. This glycocalyx shedding appears to be mediated by other sheddases than systemic tryptase release, as tryptase concentrations in blood remained unchanged during surgery.

The findings of this pilot study encourage further, larger trials to elucidate the mechanisms leading to glycocalyx shedding during oral surgery. Furthermore, approaches protecting the glycocalyx, such as normovolemia, infusion of heparanase inhibitors, steroids or doxycycline, should be investigated [7,24,26]. Protecting the glycocalyx might hold potential to accelerate perioperative recovery due to potentially improved macro- and microcirculation.

## Figures and Tables

**Figure 1 jcm-11-05797-f001:**
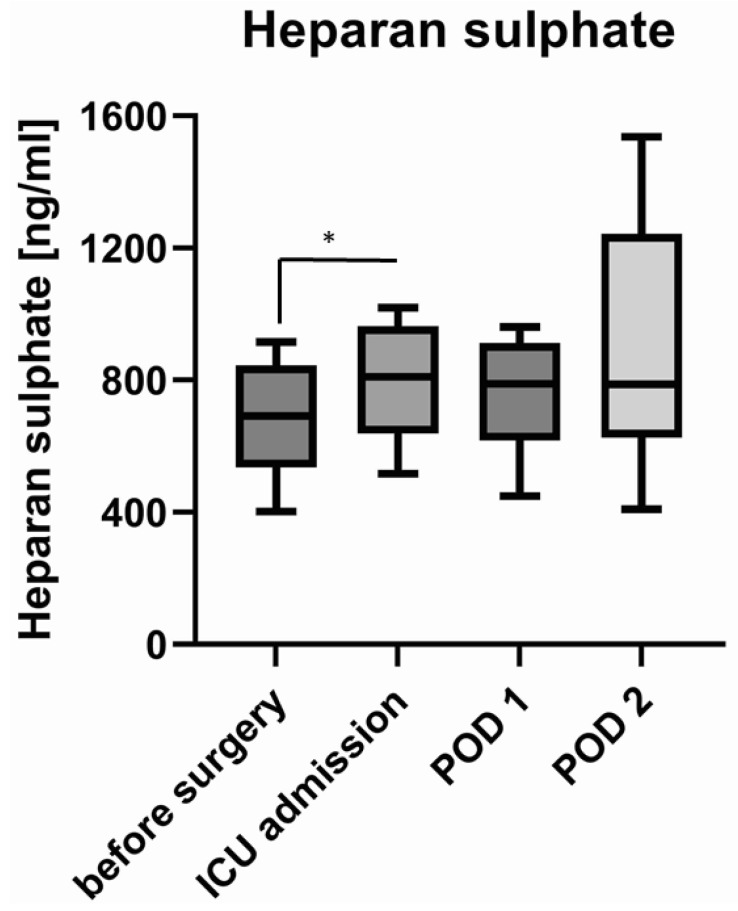
Concentration of heparan sulphate before and after surgery and on postoperative days (POD) 1 and 2. Box–Whisker Plot (Tukey method). * = *p* ≤ 0.05.

**Figure 2 jcm-11-05797-f002:**
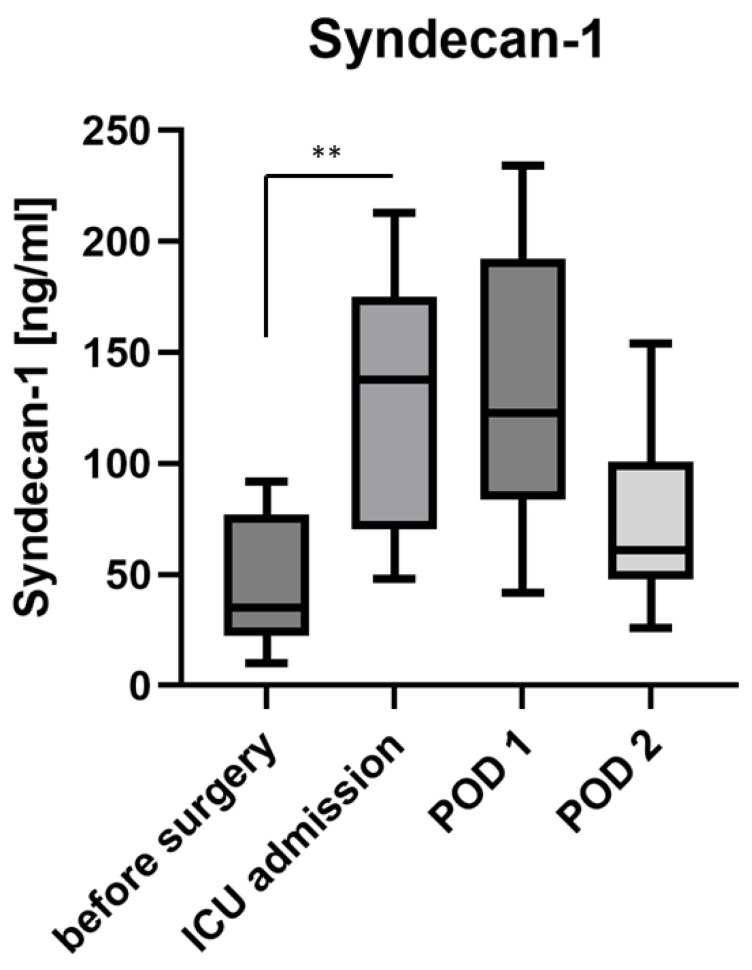
Concentration of syndecan-1 before and after surgery and on postoperative days (POD) 1 and 2. Box–Whisker Plot (Tukey method). ** = *p* ≤ 0.01.

**Figure 3 jcm-11-05797-f003:**
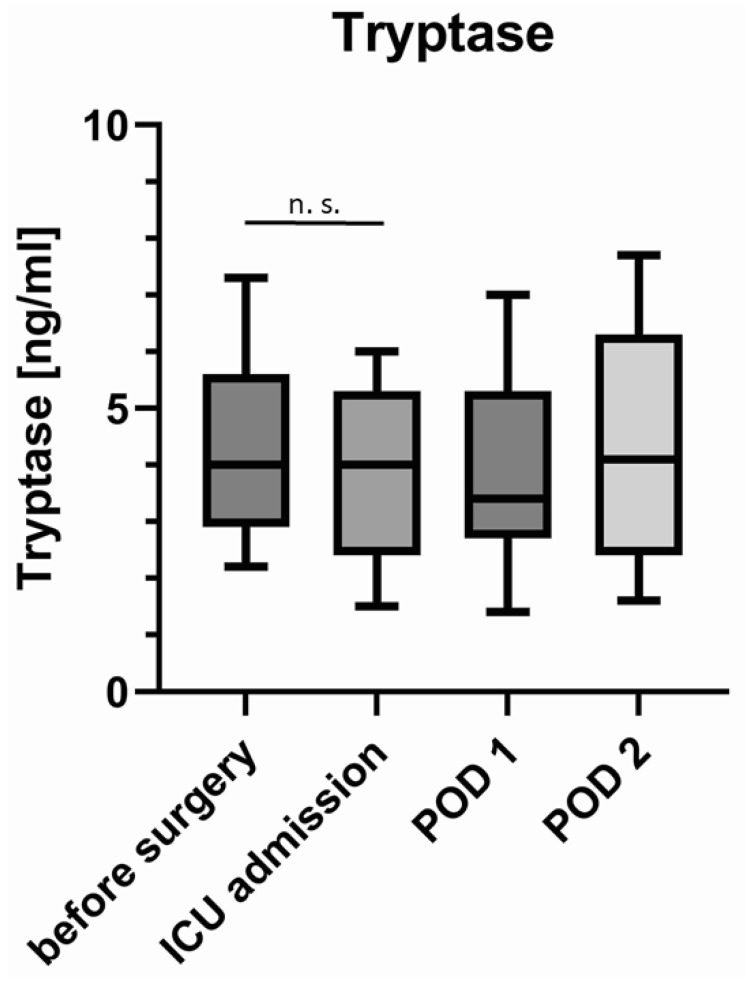
Concentration of tryptase before and after surgery and on postoperative days (POD) 1 and 2. Box–Whisker Plot (Tukey method). n. s. = *p* > 0.05.

**Table 1 jcm-11-05797-t001:** Demographic and basic clinical data of patients. Unless otherwise stated, data are presented as median (interquartile range).

Age [years]	64 (55–74)
Sex [number (%)]	female 4 (25%)/male 12 (75%)
weight [kg]	80 (65–90)
Height [m]	177 (169–180)
Diagnosis [number]	Squamous cell carcinoma of
	maxilla *n* = 2
	cheek *n* = 3
	tongue *n* = 5
	base of the mouth *n* = 3
	palate *n* = 2
	Synovial sarcoma maxilla *n* = 1
Type of surgery [number]	Radial free flap *n* = 9
	Fibular free flap *n* = 3
	Scapular free flap *n* = 1
	Latissimus dorsi free flap *n* = 1
	Anterolateral thigh flap *n* = 1
	Skin graft *n* = 1

**Table 2 jcm-11-05797-t002:** Clinical data concerning perioperative fluid balance. Data are presented as median [interquartile range].

Duration of Surgery [min]	480 (393–570)
Crystalloid infusion [mL]	5550 (4750–7000)
Colloid infusion [mL]	0 (0–500)
Red blood cell transfusion [mL]	177 (169–180)
Fresh frozen plasma transfusion [mL]	0 (0, 0)
Blood loss [mL]	1150 (650–1575)
Diuresis [mL]	1600 (1325–1893)
Maximum norepinephrine [mcg/kg/min]	0.12 (0.07–0.2)

## Data Availability

The data presented in this study can be made available upon request from the corresponding author.

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
