# Peer review of "Shedding of the Endothelial Glycocalyx Independent of Systemic Tryptase Release during Oncologic Oral Surgery: An Observational Study"

_jcm, 2022, doi:10.3390/jcm11195797_

Round 1

Reviewer 1 Report

The article reported that concentrations of the glycocalyx components heparan sulphate and syndecan-1 as well as of tryptase in blood serum before and after surgery were measured in patients undergoing oncologic oral surgery.

The major problem:

1. Too few patients were enrolled. There are no detailed inclusion and exclusion criteria and flow charts.

2. The correlation analysis between the glycocalyx shedding components and the type of perioperative fluid can be considered.

3. The current data is not sufficient to draw this conclusion, which is shedding of the glycocalyx in this case appears to be independent of mast-cell-activation and tryptase release. 

Author Response

Thank you for your critical review, which addresses important aspects of our submission.

  1. We agree that the number of patients is rather low. However, as we pointed out, previous studies (reference 10) give reason to assume that 16 patients are sufficient for a pilot study. The small number of patients is mentioned in the limitations section. The exclusion criteria are now explicitely mentioned.
  2. The correlation between fluid administration and glycocalyx shedding is interesting, indeed. We had also considered to present a correlation between fluids and glycocalyx, but decided against it, as fluid regime was not standardised and left to the discretion of the attending anaesthetist. This topic would certainly be interesting to investigate in a larger trial with standardised fluid regimes, e. g. similar to the study cited as reference no. 23 (TAPIA et al 2021).
  3. In the conclusions section, we have used a careful wording concerning our conclusion. We also mentioned that this work is a pilot study and that larger studies should corroborate the results. However, we are convinced that the way we formulate our conclusion is justified by the data we have, supported also by the additional statistical analysis without outlying values, as described in the results section

Reviewer 2 Report

Thank you for considering me as a reviewer of this manuscript. I call attention only to table 2. I believe there is an error in the values (see the "Colloid Infusion [ml] 0[0-500]") that can easily be correct.

Author Response

Thank you for your review.

In table 2, we presented median and interquartile range. That is why the median (the 8th patient of 16 ranked patients) had a colloid infusion volume of 0, i. e. received no colloids at all, and the patient at the 75th percentile received 500ml, i. e. one bag, of colloid infusion.
10 patients in our collective received no colloid infusion. The arithmetic mean would be 312.5ml and standard deviation 478.7ml. 

Reviewer 3 Report

Dear authors, the manuscript deals with an interesting subject and the way it was written can fascinate the reader of the Journal.

My suggestion is to:

- Correct some typos and the English language, especially in the discussion;

- Integrate introduction and discussion with multiple bibliographic sources;

- Learn more about the endothelial molecular mechanisms and the functions performed by the 3 substances whose levels have been measured in the blood, with the aid of images where necessary;

- Provide more details on the types of surgery your sample has undergone.

Best Regards

Author Response

Thank you for your review.

1. Several linguistic flaws have been corrected

2./3. The basics of the structure of the endothelial glycocalyx have been described with a special focus on heparan sulphate and syndecan-1. The role of tryptase is also described in the introduction. A fundamental paper about structure and function of the glycocalyx has been added as reference 1.

4. The result section has been expanded accordingly (lines 123-125)

Reviewer 4 Report

Thank you for the invitation to review this article.

I have to say that the authors did a great job. The work is presented clearly and efficiently.

Introduction is good, presents well the issue and gives background.

Study design is proper.

Results are well presented, graphically and verbally and well understood.

Discussion is good and rises fine hypotheses about the reasons behind the results accepted.

Conclusion presents the basic lack of knowledge that we have in this issue and moves readers towards further research.

Well done! 

Author Response

Thank you very much for your review.